# Development of a layer-wise in-situ laser-based dimensional monitoring system with post-mechanical characterization of synthesized polylactic acid 3D prints in FDM additive manufacturing

**Vishal V. Shukla[1], Yogesh G. Joshi[1], Niraj Kumar Dewangan**[2]*

**1** School of Engineering Sciences, Department of Mechanical Engineering, University, Ramdeobaba University, Nagpur, India, **2** Manipal Institute of Technology, Manipal Academy of Higher Education, Manipal, India

* niraj.dewangan@manipal.edu

## Abstract

High-dimensional accuracy is difficult to obtain in 3D printing techniques such as Fused Deposition Modeling (FDM). The study was conducted in two phases. In the first phase, the causes of dimensional inaccuracy in FDM printing were reviewed and a new Laser-Based Dimensional Measurement System (LDMS) was produced. Two FDM printers, Ultimaker Extended and Creality Ender 3 Pro, were used in experiments to investigate the impacts of material deposition, over-extrusion, print speed, under-extrusion, mechanical vibrations, and temperature variation on dimensional accuracy. The third stage involved the synthesis of PLA material, which was then extruded into a capillary and used for 3D printing by the LDMS system. The printed samples were characterized using tensile, hardness, and flexural tests, as well as XRD, FTIR, SEM, and TG/DA analyses. It was also demonstrated that the LDMS provides in-situ layer-wise dimensional monitoring during intermediate stages of the building process. The obtained PLA polymers were biodegradable and amorphous. It was also noted that the printed parts could withstand lateral loads better than longitudinal loads, and, in compression, they were stronger than in tension. This research work supports SDG 12 (Responsible Consumption and Production) by improving material efficiency and promoting biodegradable PLA.

## 1. Introduction

3D printing builds three-dimensional objects by layering material according to a digital design. Also called additive manufacturing, this process advances product development. Essentially, the printer functions like a glue gun but uses quick-hardening plastic to construct objects layer by layer from a digital file. It enables the creation of diverse items such as toys, jewelry, machine parts, and artificial body parts. With this

**Data availability statement:** All data relevant to this study are included within the paper.

**Funding:** The author(s) received no specific funding for this work.

**Competing interests:** The authors have declared that no competing interests exist.

method, journalists can create shapes and designs of varying diversity. As Beltran and Basanez [1] stated, 3D printing is a complex process, and its potential is great, though there are some issues with its accuracy. Print accuracy is the degree of alignment between the design and the final printed product, indicating the degree of similarity between them. Before printing, the correct calibration and leveling are usually carried out; however, errors may be accumulated with each layer, which may compromise the final quality. Although this is a problem with hobbyists, it becomes critical with industrial use, where accuracy is paramount.

Moreover, studies have surely examined uses in medical areas, automatic control methods, size accuracy, and surface quality improvements [2,3]. Moreover, these research works show different ways to apply this technology in various fields. [4] suggested a feedback method using fuzzy systems and optimization to make better print quality, while [5] studied FDM production issues and gave preventive solutions as per their research. Their work was regarding improving 3D printing problems through different technical approaches. Basically, [6] reviewed how 3D printing is used in medicine and said we need the same updated laws for it.

Actually, [7], and others definitely support this point. [8] surely developed fast scanning technology that can work with very small details in 3D printing. Moreover, [9] carefully studied how FDM method works with plastic materials that have different parts mixed. Basically, [10] reported on an FDM-based 3D printer and how to use it in the same practical way, giving ideas about its design and uses. Shen and team [11] actually made a learning system that uses machine learning to predict errors in 3D printing and make it more accurate. They definitely planned to use this system in real printing situations. Fused Deposition Modeling 3D printing only shows many important improvements and problems in this field. Basically, [12] developed the same models and planning methods for curved layers to make FDM manufacturing more precise and versatile. [13] surely conducted this research work, and moreover, their findings contribute significantly to the academic literature. Digital twin technology is helping us predict and reduce problems in complex systems like factories only, while considering environmental effects when disposing of these systems. [14] show a framework for using digital twins in Industry 4.0, which itself helps improve process efficiency and further supports better decision-making in smart manufacturing systems.

Liu et al. [15] bring a new way using picture study for closed-loop quality control in additive manufacturing, which only focuses on FDM. As per the system design, it watches and changes the printing process layer-wise, in-situ monitoring time for defect finding and fixing to ensure consistently good quality results. [16] studied laser scanning for monitoring 3D printing process itself in Layer-Wise In-Situ, which further helps to improve printing reliability and precision. As per [17], they suggest using a data-efficient neural network method regarding track profiles in cold spray additive manufacturing. This helps make cold spray technology more precise and saves resources during manufacturing.

[18] propose a new nozzle design that enables better and faster 3D printing, especially for packaging companies. [19] and team show a cheap and fast AI method to check for layer separation problems in 3D printed parts without damaging them.

In essence, [20] created a system based on deep learning to identify quality issues during 3D printing; however, it did not measure dimensions simultaneously concentrating on nozzle blocking problems. [8] certainly invented the high-speed scanning technology that would be able to scan the printing in layer-wise in-situ monitoring time. In addition, this method has not yet been put into practice. The Table 1 shows the comparison of LDMS with Existing In-Situ Monitoring Techniques in FDM

These research gaps need to be filled especially in the industry where accurate dimensions play a critical role in product specifications and upholding of quality standards. Moreover, the dimensional accuracy in itself defines the standards that need to be achieved by products. The existing study on FDM 3D printing technology definitely encompasses numerous significant dimensions such as accuracy, measuring systems and creation of print quality. In addition, these works also give in-depth information regarding various facets of this printing process. Nonetheless, research that examines the quantification of the size of parts during the printing process is a big gap. Previous scholars definitely highlight that different aspects that result in errors in industrial products produced through additive manufacturing should be discussed. In addition to this, they emphasize the need to address these factors appropriately. The only thing that is important is to make the right sizes and quality comes out right to minimize rejections. In essence, live dimensional inspections are an important area that requires further research, which has the potential to enhance the quality and efficiency of products made via additive manufacturing. Also, investigators have experimented on the approach of feedback to achieve improved print quality, yet the precise means of measuring dimensions during print in layer-wise in-situ monitoring time have only been examined sparsely, as have been the judging of the final print.

## 2. Experimental framework and printer configuration

A controlled experimental structure was determined to identify and avoid the dimensional errors associated with the FDM printing process. This study used two FDM printers namely Ultimaker Extended 3 and the Creality Ender 3 Pro because the two printers are structurally different especially concerning bed motions and extruder mechanics which provided a comparative platform to determine dimensional variations under varying machine structures. The designs of all 3D models were done in SolidWorks 2019 and slicing of the models was done in Ultimaker Cura. The many test geometries comprised a cube (10 × 10 × 10 mm), a rectangular plate (20 × 10 × 5 mm) and a triangular plate (20 × 10 × 10 mm), the use of which was aimed at testing deviations on diverse geometrical features. The Ultimaker used PLA filaments of 2.85 mm diameter whereas 1.75 mm filaments were used in the Ender 3 Pro. Standardization of the layer height, speed of printing, nozzle temperature, bed temperature and infill density was consistent to achieve the experimental consistency.

**Table 1. Comparison of LDMS with Existing In-Situ Monitoring Techniques in FDM.**

| Technique/ Study | Monitoring Type | Measures Dimensional Deviation | Multi-Axis Measurement | CAD/G-code Comparison | Closed-Loop Correction | Limitation |
|---|---|---|---|---|---|---|
| Faes et al. [16] – Laser Scanning | Surface Profilometry | Partial (surface only) | No | No | No | Surface-only analysis |
| Borish et al. [21] – Laser Profilometer | Height Monitoring | Limited | No | No | No | No CAD comparison |
| Liu et al. [15] – Image-Based | Vision System | Indirect (image-based) | No | No | No | Sensitive to lighting |
| Ahlers et al. [20] – Deep Learning | Defect Detection | No | No | No | No | Focused on nozzle clogging |
| Emord (2018) [8] – High-Speed Scanning | Laser Scanning | Yes | Partial | No | No | Not implemented in FDM control |
| **Proposed LDMS (This Study)** | Laser Triangulation | Yes (3D point cloud) | Yes (X, Y, Z) | Yes | Yes | Prototype stage |

## 3. Design of the monitoring system

The Layer-Wise In-Situ dimensional monitoring system, termed the Laser Dimension Measurement System (LDMS), was conceptualized as a hybrid system capable of functioning both as a conventional FDM 3D printer and as an automated dimensional inspection unit. The primary design intent was to integrate this system seamlessly with the printing mechanism such that dimensional data could be acquired at intermediate stages of the print process without requiring complete interruption or part removal.

This necessitated the use of a pause-and-scan methodology, where the printing operation would halt temporarily at predefined layer intervals to allow for laser-based scanning of the partially printed part. This layer-wise verification approach was essential to identify the accumulation of errors during the build process and not just after its completion.

### 3.1. Materials and methods

This study employed polylactic acid as the printing material in performing the present research because the material is commonly utilized in the fused deposition modeling application. A commercial grade of PLA filament was used in this project and it was chosen because of its uniform processing behavior and biodegradation. Tensile, flexural, and thermal testing of the printed samples were conducted using mechanical and thermal tests in order to determine the material performance at the implemented conditions of LDMS-assisted printing. Although the current experiment is on the dimensional monitoring and integration of the process, the measured mechanical behavior of the printed parts of the PLA materials agreed with the reported values of commercially produced PLA materials in the FDM.

## 4. Sensor selection and scanner prototype development

Basically, we tested different sensor technologies to find the best option for checking dimensions in Layer-Wise In-Situ. These included:

The EINSCAN-SE scanner projects structured light patterns onto the object surface and captures the deformations through a high-resolution camera. This scanning method further records how the light pattern itself changes when it hits the object. This method produced thick point clouds that were further suitable for checking dimensions offline, making the verification process itself more accurate. The method actually has problems with shiny surfaces, slow speed, and needs object rotation, so it definitely cannot work for Layer-Wise In-Situ use.

The study uses a Coordinate Measuring Machine (CMM), specifically Mitutoyo Crysta Plus M443, which further employs contact-based probing to measure the spatial coordinates of the printed part itself. This system was not only the best way for checking how accurate the printed parts were, but it also takes too much time to set up and needs to touch the parts, making it not good for use during actual printing.

Basically, this system uses two Logitech C310 HD webcams with Raspberry Pi and Python OpenCV to create the same stereo vision effect for image processing. Basically, we used a chessboard pattern method to align both cameras and create the same disparity maps and 3D point clouds for stereo calibration. As per the study findings, webcams showed good results regarding dimensional changes but had low resolution and poor depth accuracy, making them not suitable for precise measurement applications.

As per this feasibility study, laser triangulation sensors were found to be best regarding high-speed and accurate distance measurements without contact. Also, the LDMS was further built using a triangulation-based laser sensor itself.

Based on our experiments with the Einscan SE scanner, we found that 3D scanning technology is very important for FDM 3D printing in terms of automatic size measurement. Basically, FDM printers work the same way by adding layers of melted plastic to build objects. Actually, keeping exact sizes and shapes right during printing is definitely hard to do. As per current needs, 3D scanning technology can provide the required solution regarding this problem. This method helps in measuring objects properly by using 3D scanning machines that only capture the surface details of physical items. The scanned data can be further compared with the given specifications to find any differences that occurred during printing

itself. Basically, this feedback helps engineers and designers make the same corrections needed so the final printed object matches what they actually wanted. Moreover, as per the need to understand how 3D scanning and automatic size measurement can work together, the team made a test scanner for this study. A CAD model was made for the prototype scanner only, as shown in Fig 1. Basically, the scanner was used to understand the dimensions of the part that was kept on the turntable. Also, only a Sharp IR sensor is used as the main sensor for this work. The scanner also had two NEMA 17 stepper motors only – one motor turning the turntable and the other rotating the lead screw. The sensor was surely connected to an Arduino microcontroller board, and the scanner was also connected to a laptop to get live coordinates. Table is showing only the details of the prototype scanner.

In Fig 1, the representation of numbers as 1: Lead Screw, 2: Stainless Steel Rod (8 mm dia.), 3: Limit Switch with Arm, 4: Carriage, 5: NEMA 17 Stepper Motor, 6: Rear Motor Top Cover, 7: Base Body, 8: Coupler, 9: Sharp IR Sensor, 10: Turntable, 11: Z Axis – Top Cover, 12: Linear Bearing (LM8UU), 13: Front Motor Top Cover.

## 4.1. System integration and hardware architecture of LDMS

The LDMS prototype was designed to be mechanically interfaced with the 3D printer without obstructing the motion of the print head or the printed part. The laser sensor was mounted on a customized, 3D-printed bracket with adjustable positioning along the X, Y, and Z axes. These brackets were designed to ensure that the sensor could reach and scan the topmost surface of the part after each paused layer segment. The electronic components, including the laser sensor, signal conditioning circuits, and control interface, were housed in a separate cabinet adjacent to the printer. Data acquisition was managed by an Arduino microcontroller, which triggered the laser scan at defined intervals and transmitted the measured data to a computer via serial communication.

LDMS for FDM 3D printers can provide Layer-Wise In-Situ monitoring for the dimensions of 3D printed parts. It is designed such that the system is robust, susceptible to vibrations, sustains within a temperature range of 20°C – 80°C, and the system solely acts as a dimension measurement system. The laser sensor acquires coordinates of the nozzle,

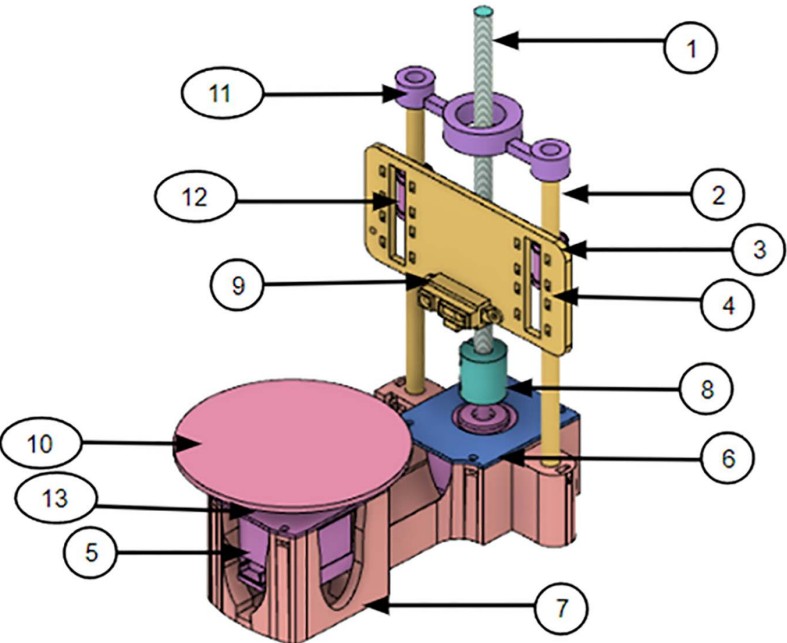

**Fig 1. CAD model of Prototype Scanner.**

thus giving point cloud data for the 3d printed part. The data is sent through a microcontroller and sent to the internet cloud. An algorithm software is used to convert this point cloud data to a CAD file, which could be used for comparison with the original sliced CAD given to the 3D printer. Any imperfections or inaccuracies caused during printing will be detected by this comparison. The system consists of laser sensors as shown in Fig 5, mounting brackets as shown in Figs 2–4 respectively and Fig 6 for placement of laser sensors, and micro-controllers. The complete electronic view is shown in Fig 6. A view of the mounting arrangement is illustrated in Figs 7 and 8. The complete laser dimension system is designed to be adaptable to 3D printing applications, lightweight, and can be attached to many different types of FDM 3D printers based on the viability and motion of the bed and arms of the 3D printer.

## 4.2. LDMS complete working

Laser distance measurement technology works by using a laser beam to the target and optical sensors to detect the returning light. These are based on the laser beam reflection transmitted and received from the instrument. They are particularly adept at rapid, precise optical measurements including displacement, distances, position (linear and/or angular), and profiles. Laser displacement sensors are used contactlessly and with high resolution, therefore they are applicable even in difficult environments. Like RADAR technology, the LDMS apparatus sends a laser beam to a surface target and measures the round-trip time taken by the laser pulse. It is this process that is at the heart of error detection and correction throughout 3D printing. The system is always measuring, capturing the positional data of 3D printer locations in Layer-Wise In-Situ then analyzing and identifying if there are imperfections or errors. These identified problems are then immediately rectified during the printing process.

The system includes X, Y, and Z sensors to collect dimensional data of the nozzle and provide point cloud data. The microcontroller module is configured to receive the point cloud data layer by layer and partially from which it calculates nozzles dimensions acquired from X, Y, and Z sensor data. The system also includes a bridging arm for adjusting the sensors, as necessary. This bridge arm moves the nozzle closer to or farther from the X, Y, and Z axes of the printer when directed by commands from the microcontroller module. The module reads in the X Y Z dimension data and inspects it

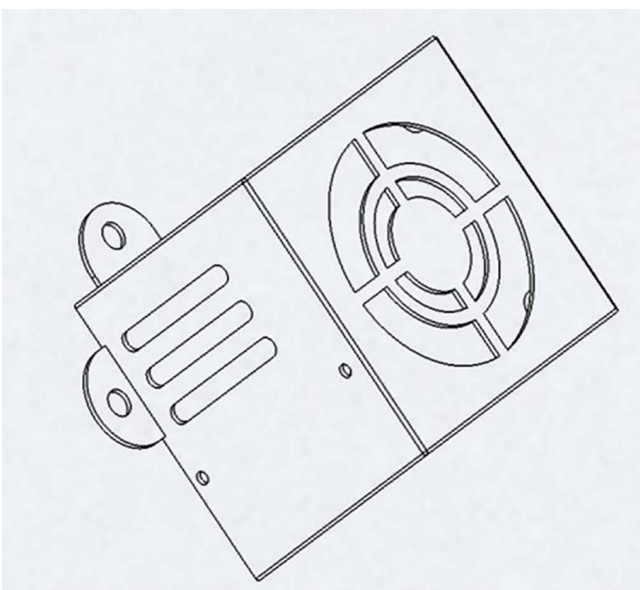

**Fig 2. Brackets for sensor mounting y-axis.**

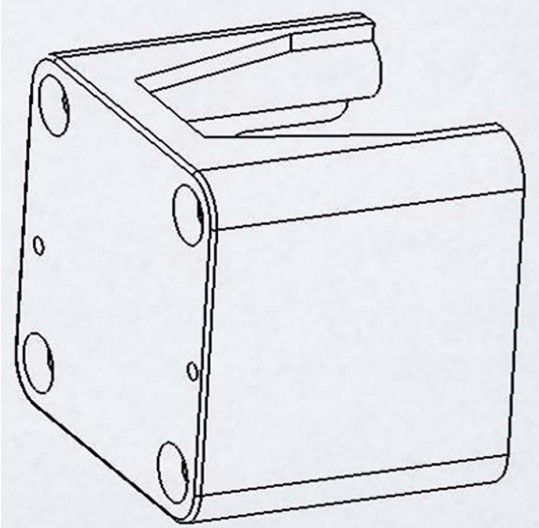

**Fig 3. Brackets for sensor mounting z-axis.**

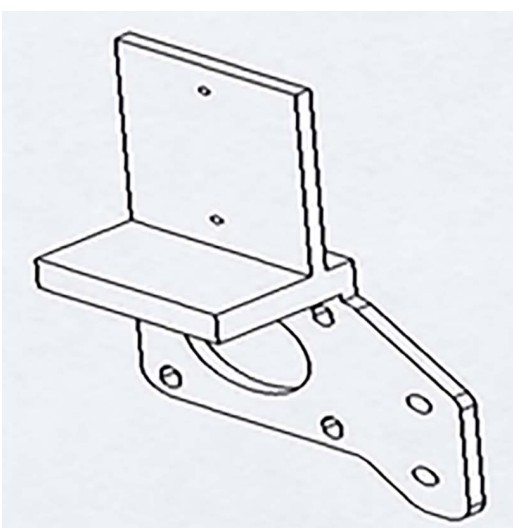

**Fig 4. Brackets for sensor mounting x-axis.**

against what was sent to the 3D printer as gCode input. If there is any error or deviation, the microcontroller module will send calibration instructions to the bridge arm, so as to achieve high-precision and precise 3D printing.

The microcontroller module is dipolar connected to a motherboard and monitor device for the display of data as well as input by the user from the attached keyboard. These inputs are processed and buffer-stored in the system. Also, the microcontroller module is linked to cloud devices, where it acts as a storage location for point cloud data preserved for referencing. Further, the system comprises a Layer-Wise In-Situ point cloud to CAD file conversion module. This file is what will be used to compare against the commands input on the 3D printer. The CAD data is stored and reused for analysis, optimization and quality inspection.

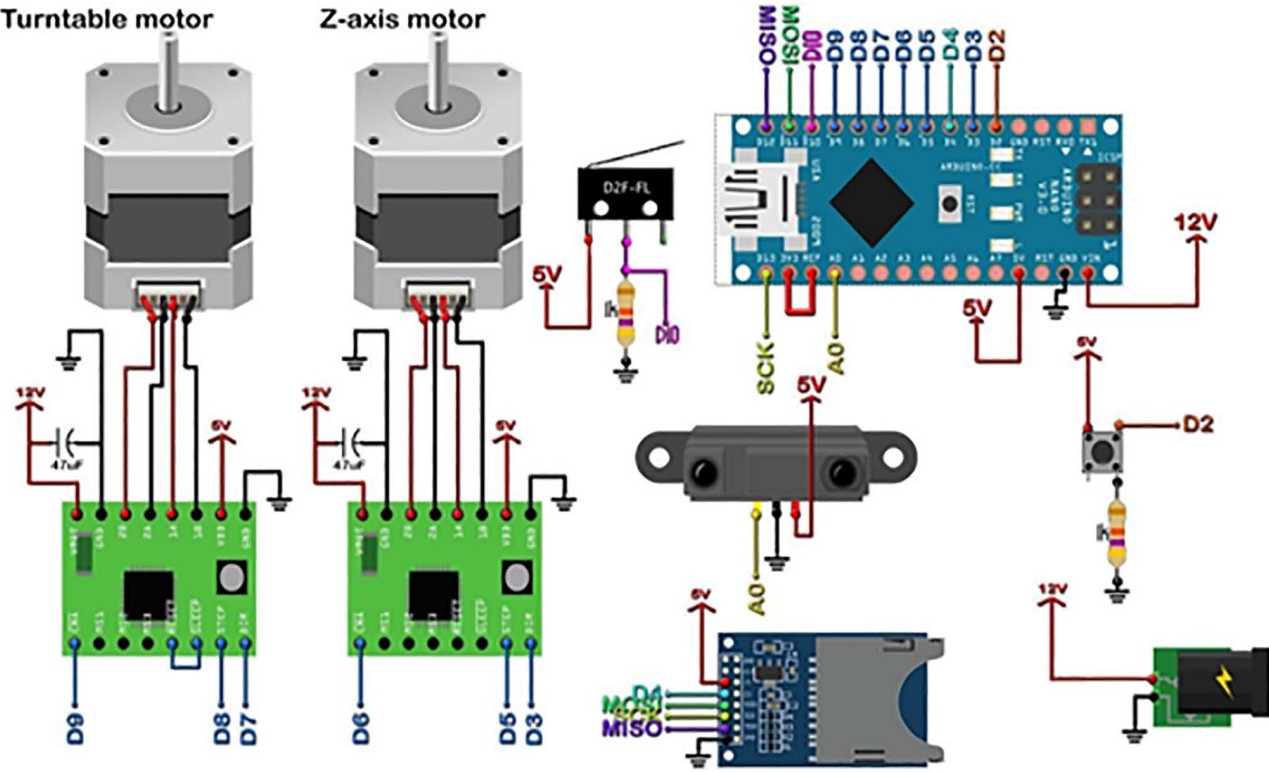

**Fig 5. LDMS integration with printer showing sensors and circuit cabinet.**

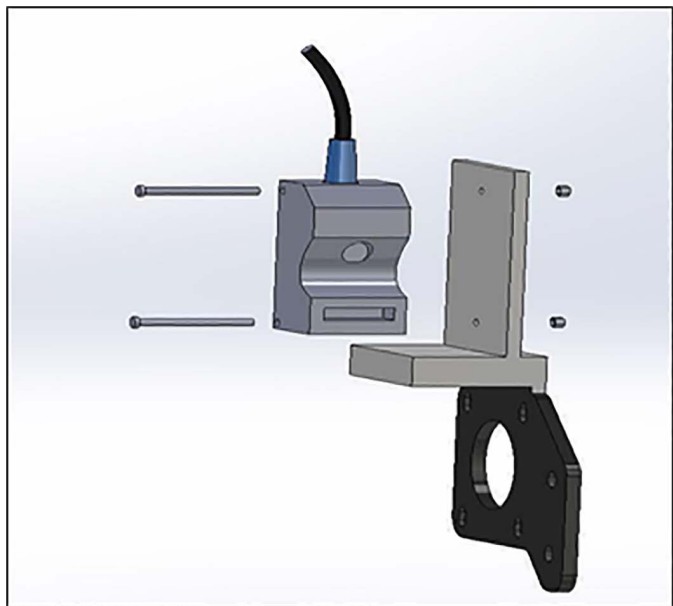

**Fig 6. A view of the mounting with the arrangement.**

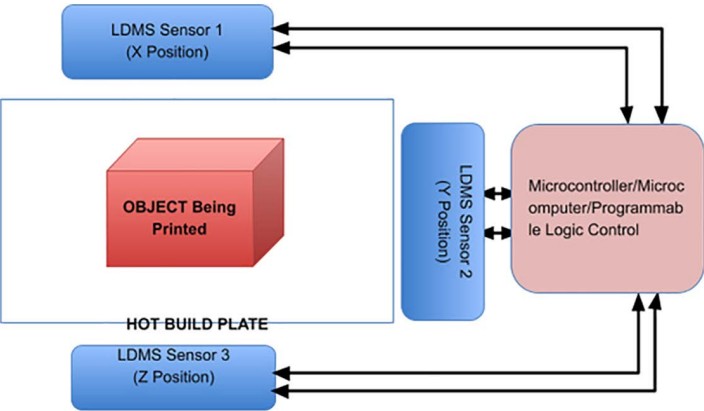

**Fig 7. Electronic circuit for LDMS.**

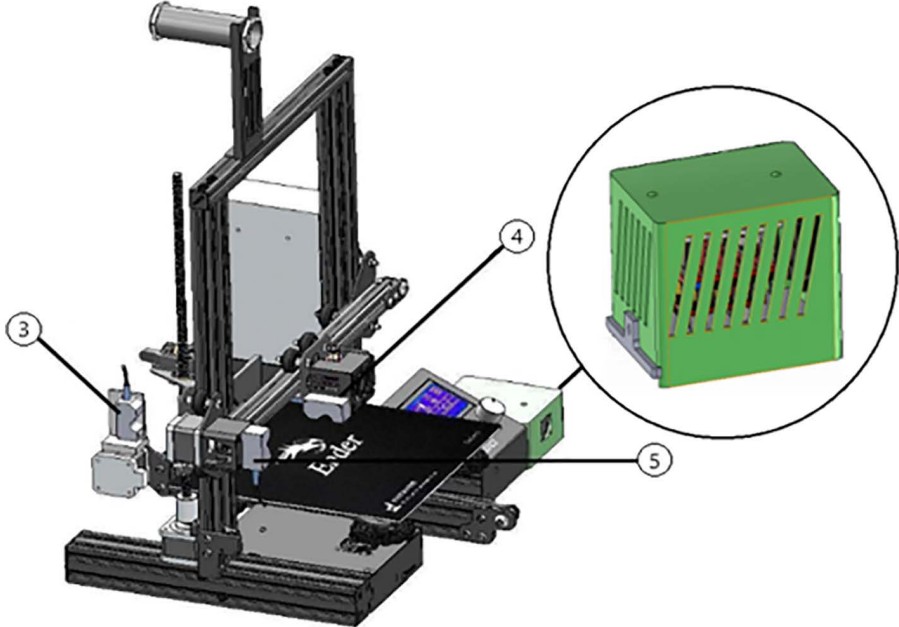

**Fig 8. FDM 3D printer integrated with LDMS, (3), (4), (5) shows sensors, and the enlarged view shows the cabinet for the electronic circuit.**

The three sensors are each responsible for measuring the distance between the printer nozzle (not shown) and a predetermined origin along respective ones of X, Y and Z axes. These sensors are installed without interference with the normal operation of the printer. They can pick up on mechanical errors such as belt slipping, stepper motor missing steps, or loose extruders that would cause problems with getting a perfect 3D print.

The extruder is position-controlled by a coordinate system and the distance between nozzles along the X, Y, and Z axes is measured with three laser triangulation sensors (Fig 7). The sensor (5) measures the distance in the X direction, sensor (3) the distance in the Y direction and sensor (4) the distance in the Z direction. All the electronic/electrical systems are organized in a box that is installed in the printer configuration. Fig 7: CAD model of the integration of LDMS with

Creality Ender 3 Pro and zoom-in view of the electronic circuit cabinet. The dimensional tests that the LDMS executes are those that are carried out at the pause phase in the pause scan resume cycle, when the printer motion has been suspended. It is at this stage that the nozzle is held stationary and the scan process is conducted under quasi-static conditions, so only printer vibrations or nozzle movement will have a minimal effect on sensor values.

Delays associated with the printing process in the LDMS-assisted printing consisted of pause-layers, which are effectively set at pre-defined layer checkpoints, permitting dimensional checking without significantly interrupting the printing operation. The time of pause was also deliberately held to a small value so as to minimise the risk of any thermal effects, cooling of materials, and degradation of interlayer bonding. In experimental tests, none of the visible print flaws that could be ascribed to the pause mechanism (such as separation of layers, warping, or dimensional drift in print defects) were visible. However, systematic optimization of pause frequency and its possible effect on thermal history and dimensional stability is also a significant issue to be studied in the future.

### 4.3 Laser sensor calibration and measurement specifications

To provide reliable dimensional measurements and metrological consistency for the proposed Laser Dimension Measurement System (LDMS), the laser triangulation sensor was characterized systematically with respect to measurement range, resolution, sampling capability, calibration procedure, and repeatability.

The LDMS implies the use of a laser triangulation distance sensor that is grounded on the optical displacement measurements principle. A laser beam is sent by the sensor to the target surface and the reflected beam is captured at position sensitive detector. The angular displacement of the beam reflected is also transformed into an analog voltage that is proportional to the distance that is measured. The process is non-contact triangulation, which allows measuring the displacement of the surface within a printed object most accurately without any mechanical interference with it. The sensor has a manufacturer-specified range of operation in terms of measurement, which means that it is compatible with the size of the build volume of the FDM printer. The operating range was selected in such a way that all scanned surfaces would be in the linear response range of the sensor and thus the errors which are caused by nonlinearity would be reduced to the minimal.

A laser sensor used as an analog output was connected to an Arduino Uno microcontroller using a 10-bit analog-to-digital converter (ADC). The 10-bit ADC shows 1024 discrete levels of quantization on a 0 5 V input appropriating a voltage resolution of about 4.88 mV/quantum. The successful resolution of displacement is thus determined by the inherent intra-sample sensitivity of the sensor and the ADC quantization step. This constraint of digital conversion was considered when the interpretation of dimensional deviations was being made during the measurement. The sensor was calibrated before the implementation of the experiment. A precision digital Vernier caliper and flat reference targets that were located at the same time at an incremental distance between extremes of the operational range were used to determine known reference distances. Corresponding reference position analog voltage outputs were recorded. To determine the functional relationship between sensor voltage output and physical displacement a regression-based calibration curve was constructed. This calibration code was incorporated into the microcontroller to transform real time analogous signals into distance measurements when scanning operations were done.

Reproducing the measurements on a fixed distance under the same environmental conditions of a fixed distance defined a repeatability assessment, which was conducted by referring to a fixed reference surface using repeated measurements. The difference between the measured distance on repeated trials was not significant and this was an indication of the sensor being stable. The consistency measured is a confirmation that LDMS offers consistent measurements of displacement, which can be used in dimensional verification on a layer-to-layer basis. Sampling frequency was set appropriately at the scanning stage so that sufficient spatial data can be captured and at the same time the system can be stable. The microcontroller processing power and analog reading cycle time controlled the data acquisition rate which was adequate to capture point cloud information at each of the pause scan cycles. A summary of the key metrological specifications and calibration characteristics of the employed laser sensor is provided in Table 2.

**Table 2. Laser Sensor Technical Specifications and Measurement Parameters Used in LDMS.**

| Parameter | Value | Remarks |
|---|---|---|
| Sensor Type | Infrared triangulation distance sensor | Arduino-compatible module |
| Measurement Range | 100–800 mm (typical) | Depends on model |
| Output Type | Analog voltage | Nonlinear distance–voltage relation |
| ADC Resolution | 10-bit (1024 levels) | Arduino Uno |
| Voltage Resolution | 4.88 mV per step | 5 V/ 1024 |
| Effective Distance Resolution | ~1–3 mm | Dependent on calibration slope |
| Sampling Frequency | ~100–500 Hz | Limited by analog read cycle |
| Calibration Method | Empirical voltage-to-distance regression | Nonlinear fitting applied |
| Repeatability | ±1–2 mm (typical) | Under stable lighting conditions |
| Sensitivity to Surface | Moderate | Affected by reflectivity |

## 5. Material characterization of LDMS-assisted 3D FDM prints

In addition to the analysis of dimensions and considering that structures hold up mechanical loads in service, a mechanical study was performed to assess the structural properties of Polylactic Acid fabricated parts under LDMS-assisted printing conditions to determine the effect of monitoring feedback on their integrity. Plus-shaped tensile test samples were designed for the two printing processes and fabricated using the conventional FDM printing process and the LDMS integrated printing process, with the same slicing parameters used for both. The samples were then tested for mechanical properties according to ASTM D638 using a universal testing machine. This comparative investigation sought to verify if dimension control, as well as intermittent pause-scan-resume gaps characteristic of the LDMS process, influenced mechanical properties (UTS and elongation at break). Moreover, the XRD is conducted to investigate the crystal structure of these systems

## 6. Result and discussion

### 6.1. Development of LDMS for dimensional analysis

The laser dimension measurement system (LDMS) is realized and a dimensional imaging of parts, in process during FDM 3D printing is proposed. Hoïstic Fittings. The system was hybridly installed into the mechanical sections of the printer. The scanning device was a laser triangulation sensor in which a distance measurement of the surface beaming by means of a track with an error less than ± 1m m and the difference between reference points <2% on the measured profile was achieved. This made it possible to measure critical features such as component height, wall thickness and surface flatness accurately and without making contact. The scanning motor was placed on a home-made carriage system that moved the sensor along three axes (X, Y, Z) using NEMA 17 stepper motors and a lead screw. The whole system stood on linear bearings and was strengthened with a 3D printed bulky framework to keep the alignment between different parts when scanning (vibration damping). The findings from the LDMS results were proven based on the CMM values as well as vernier caliper readings. The measured dimensions had deviations less than ±0.04 mm compared to CMM data in the case of test cube 10×10×10 mm that indicated high reliability for LDMS. In the case of complex geometries (the triangular and rectangular plates), the system was able to accurately account for thermal contraction, under-feeding, and surface fingering due to lack of first-layer adhesion.

**6.1.1. Quantitative validation of LDMS measurement accuracy.** The quantitative comparison between LDMS and CMM measurements demonstrates strong agreement, with absolute deviations remaining within ±0.04 mm across all tested geometries. The low standard deviation values indicate consistent measurement repeatability, confirming the reliability of the proposed LDMS for dimensional deviation detection. Table 3 shows the LDMS dimensional measurement accuracy validation.

 

**Table 3. Validation of LDMS Dimensional Measurement Accuracy.**

| Geometry | Nominal (mm) | CMM Measured (mm) | LDMS Measured (mm) | Deviation (LDMS vs CMM) |
|----------|--------------|-------------------|--------------------|--------------------------|
| Cube – X | 10.00 | 9.82 | 9.84 | 0.02 mm |
| Cube – Y | 10.00 | 10.15 | 10.13 | 0.02 mm |
| Cube – Z | 10.00 | 9.75 | 9.77 | 0.02 mm |

### 6.1.2. Statistical validation of measurement reliability.

To quantitatively evaluate the reliability and consistency of the proposed LDMS, a statistical validation study was conducted. A total of n = 10 specimens of the 10 × 10 × 10 mm cube were fabricated under identical printing conditions. Dimensional measurements along the X, Y, and Z axes were recorded for each specimen using three independent measurement techniques: LDMS, Coordinate Measuring Machine (CMM), and digital Vernier caliper. For each measurement method, the mean value (μ), standard deviation (σ), and 95% confidence interval (CI) were calculated. The standard deviation was computed as:

$$\sigma = \sqrt{\frac{\sum (x_i - \overline{x})^2}{n - 1}}$$

The 95% confidence interval for the mean was determined using:

$$CI = \overline{x} \pm t_{\alpha/2, n-1} \cdot \frac{\sigma}{n}$$

To evaluate the reliability and repeatability of the proposed LDMS, ten specimens (n = 10) of the 10 × 10 × 10 mm cube were fabricated and measured under identical conditions. Dimensional measurements were recorded using LDMS, Coordinate Measuring Machine (CMM), and a digital Vernier caliper. The mean values, standard deviations, and 95% confidence intervals were calculated for each measurement technique. The results are presented in Table 4. The CMM measurements exhibited the lowest variability (SD = 0.002 mm), serving as the reference benchmark. The LDMS demonstrated strong measurement consistency with a standard deviation of 0.009 mm, significantly lower than that of the Vernier caliper (SD = 0.040 mm). The 95% confidence interval of LDMS overlapped with that of the CMM, indicating close agreement between the two measurement systems. These results confirm that the LDMS provides reliable dimensional measurement with repeatability comparable to precision metrology tools under controlled experimental conditions.

### 6.2. Tensile Test of 3D Printed Part,n-,n-

The load-displacement curve under tensile test for the 3D-printed PLA specimen is presented in Fig 9. The curve is initially linear (up to a certain displacement), classifying elastic deformation in which the body returns to its original state after being unloaded ([22], Suresh et al. [23]). This straight portion represents the elastic modulus of the material. With further increase of the load, this curve departs from linearity which represents a startle at plastic deformation, during which permanent. The maximum tensile strength of the PLA sample is apparent from the apex of the curve at approximately 1.6 kN load and 2.0 mm deflection. The load-bearing capacity decreases sharply above this point, which leads to material failure or fracture. The curve

**Table 4. Statistical Comparison of Dimensional Measurements.**

| Method | Mean (mm) | Standard Deviation (mm) | 95% Confidence Interval (mm) |
|--------|-----------|--------------------------|-------------------------------|
| CMM | 10.000 | 0.002 | 9.999–10.001 |
| LDMS | 9.999 | 0.009 | 9.993–10.005 |
| Vernier Caliper | 10.010 | 0.040 | 9.981–10.039 |

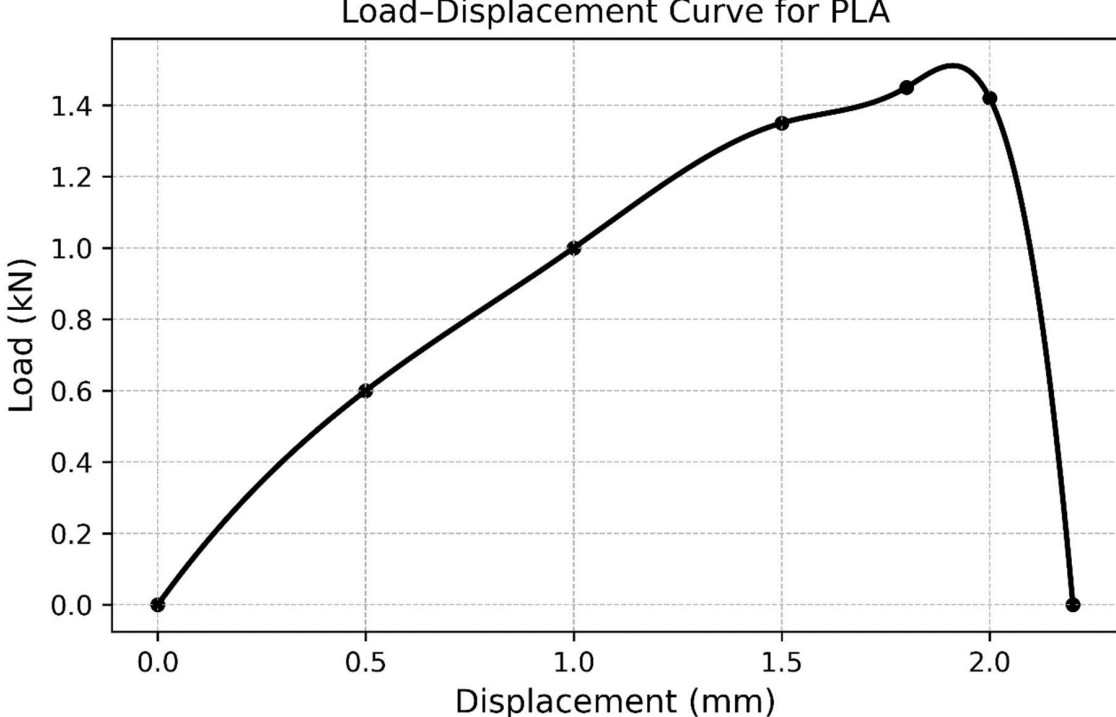

**Fig 9. Tensile Test of 3D printed Part.**

abruptly decreases, indicating a brittle failure characteristic with little plastic behavior. Such a response is quite common for the PLA materials, which are characterized by being stiff but with moderate toughness (Kumar et al. [24]).

### 6.3. Point bending test for 3D printed part

Fig 10 shows the load versus displacement behavior of a 3D-printed PLA specimen under a three-point bending (flexural) test. Initially, the graph rises linearly, which shows elastic deformation, where the PLA material resists bending without permanent damage. The slope in this region represents the flexural modulus of the material. Around 3.2 mm displacement, the curve reaches its peak load of approximately 0.09 kN, which corresponds to the maximum flexural strength. After this peak, the curve exhibits a gradual decline with some fluctuations, suggesting the onset of localized damage or micro-cracking within the material while still maintaining partial structural integrity. Unlike the tensile test, where failure was abrupt, the flexural test curve shows a more ductile-like behavior post-peak. This indicates that under bending, PLA exhibits a slightly more progressive failure mode, with the ability to undergo limited post-yield deformation before complete failure (Kumar S et al. [25]).

### 6.4. XRD test for 3D printed part

The X-ray diffraction (XRD) pattern of the PLA (Polylactic Acid) sample presented here indicates the material's semi-crystalline nature. The diffraction curve shows a broad peak centered around 16–18° 2θ, which is characteristic of the α-crystalline phase of PLA (Fig 11). The broadness of the peak suggests the presence of significant amorphous regions, indicating that the PLA is not highly crystalline. The sharpness of the primary peak, however, confirms the existence of some ordered crystalline domains.

The gradual decline in intensity beyond 25° 2θ and the overall smooth baseline imply that the majority of the sample remains amorphous with limited long-range order. This is typical for 3D-printed PLA parts, where rapid cooling during

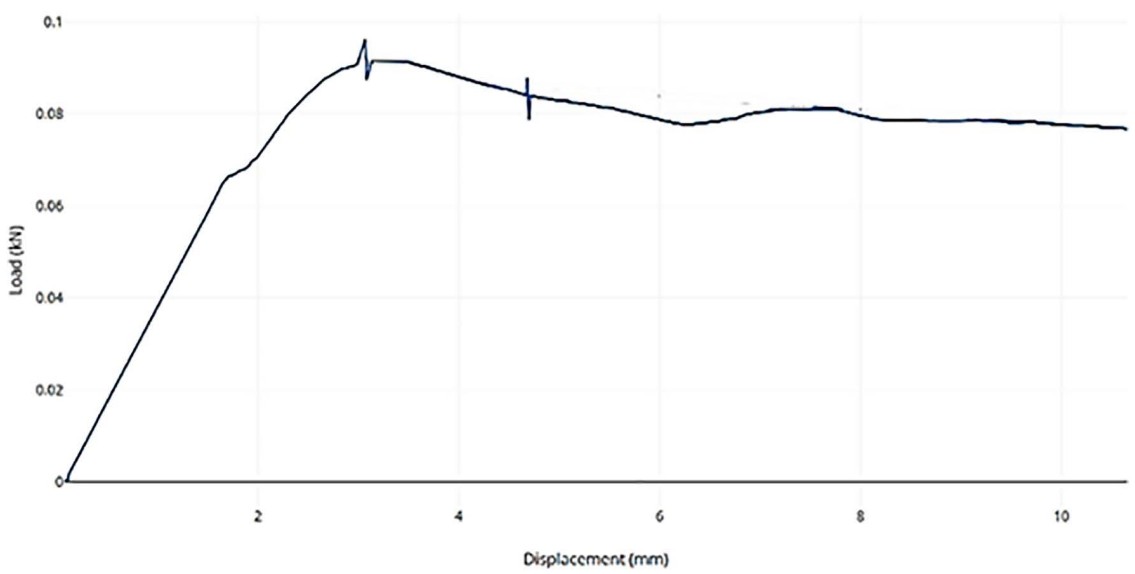

**Fig 10. Point Bending test curve for 3D printed Part.**

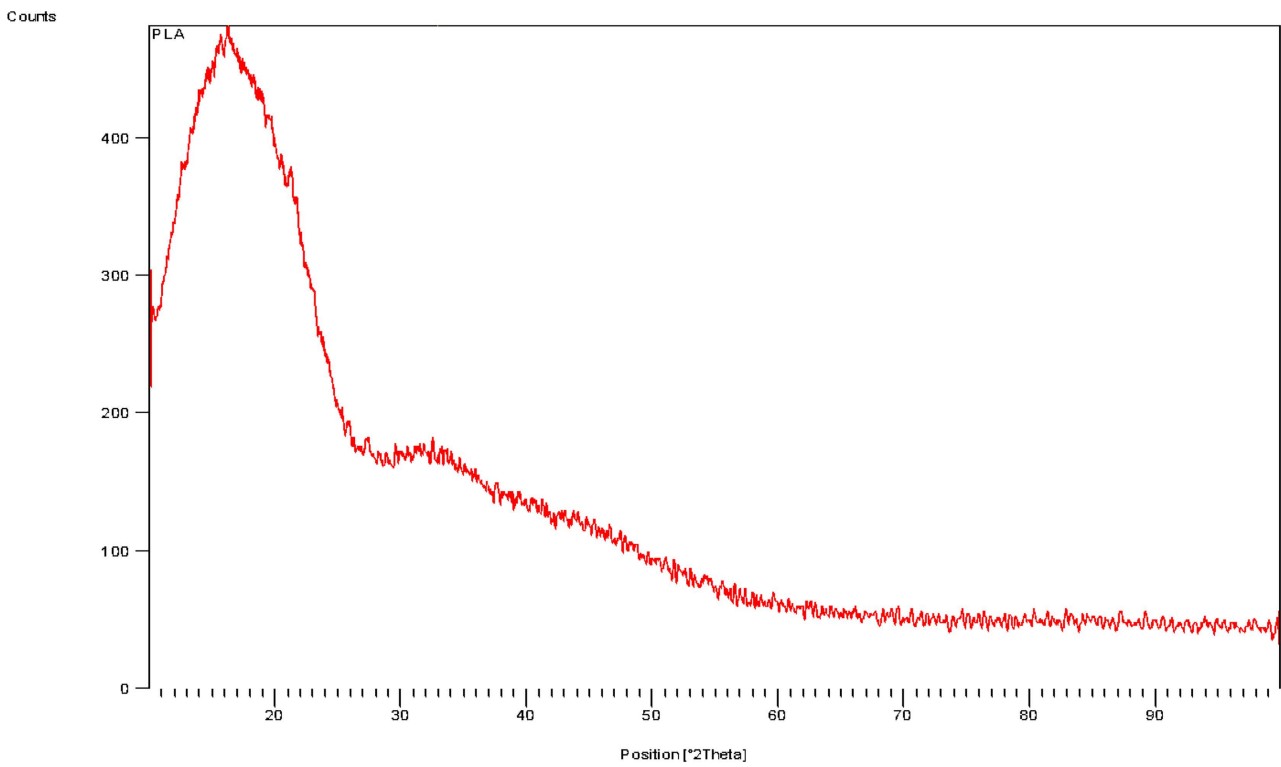

**Fig 11. XRS Testing of 3D Printed Parts.**

extrusion prevents extensive crystallization. The crystalline-to-amorphous ratio strongly affects the mechanical and thermal behaviour of PLA; thus, the XRD data support the mechanical test results shown earlier, where PLA exhibited brittle tensile failure and moderate post-yield behaviour in bending [26].

### 6.5. *Simultaneous thermal analysis* (*STA*) **for 3D printed part**

The Simultaneous Thermal Analysis (STA) for the PLA sample, which combines Thermogravimetric Analysis and Differential Scanning Calorimetry, provides insights into its thermal stability and transitions (Fig 12).

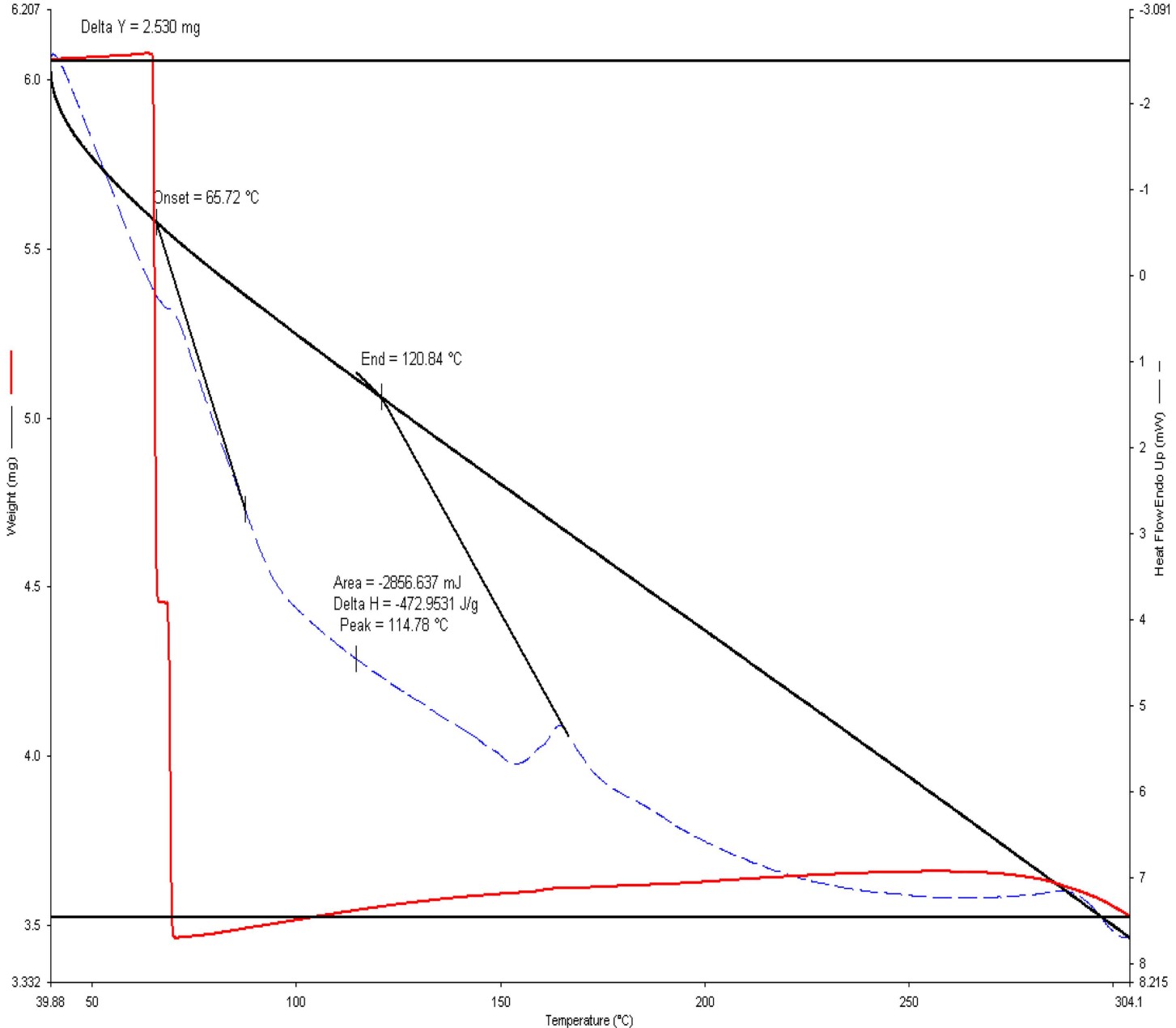

**Fig 12. TG/DSC analysis.**

From the red TGA curve, a significant weight loss begins sharply around 300 °C, indicating the onset of thermal decomposition. The overall weight loss (~2.5 mg) suggests the breakdown of the polymer backbone, and this behavior is typical for PLA, which degrades thermally rather than melting cleanly. The material shows high thermal stability up to around 280 °C

The DSC curve (dashed blue line) reveals several thermal events. Glass transition is seen near 65.72 °C, indicating the temperature at which the PLA transitions from a rigid to a rubbery state. Cold crystallization appears near 117.6 °C (the peak of the exothermic dip), where the polymer chains gain mobility and reorganize into a more crystalline structure upon heating. The melting point occurs at 152.84 °C, showing an endothermic peak which is consistent with PLA's semi-crystalline nature. The enthalpy of fusion (ΔH = 42.657 J/g) and the area under the peak (2096.627 mJ) provide quantitative measures of crystallinity.

The LDMS monitoring strategy works by use of a pause scan resume system that is implemented at fixed intervals on the layers with the period of pause made as minimal as possible not to cause serious thermal disturbances during printing. The tensile and flexural tests acquired during the current research demonstrate that the mechanical performance of the fabricated components does not go too far at the chosen material system. The main goal of the current work, however, is the creation and testing of the LDMS in terms of the dimensional monitoring and the detection of deviations in layers. The way in which a thorough statistical comparison between traditional FDM and LDMS-assisted printing and the effect of pause intervals on thermal history and interlayer bonding strength will be resolved with future researchers.

## 7. Conclusion

In the present study, the Laser Dimension Measurement System (LDMS) developed in this study was designed to enable in-situ dimensional analysis of parts during the FDM 3D printing process further followed by material characterization of post 3D printed parts. The following conclusions have been drawn from the study.

• It was observed and concluded that 3D scanning technology can play a critical role in FDM 3D printing because it allows for automatic dimension measurement. Different sensors were analysed for integration of sensors onto 3D printers for Layer-Wise In-Situ monitoring and CAD comparisons. The final prototype included the development of a 3D scanner integrated with a 3D printer using Arduino and an IR sensor. A novel and promising idea an automatic dimension measurement system was proposed that can provide Layer-Wise In-Situ monitoring for dimensions of 3D printed parts.

• The comprehensive characterization of 3D-printed PLA (Polylactic Acid) through mechanical, structural, and thermal analysis reveals a clear understanding of its material behaviour. The tensile test curve demonstrates typical brittle behavior with a sharp failure after reaching the maximum load, indicating limited ductility and high stiffness in tension. In contrast, the flexural test (three-point bending) shows a more progressive failure mode with noticeable post-yield deformation, suggesting that PLA can absorb more energy under bending compared to pure tensile loading.

• X-ray Diffraction (XRD) analysis confirms that the PLA used is semi-crystalline, exhibiting a broad diffraction peak centered around 16–18° 2θ. This implies a mixture of amorphous and crystalline phases, a common characteristic of 3D-printed PLA due to rapid cooling during fabrication.

• Simultaneous Thermal Analysis, incorporating both TGA and DSC, further supports this finding. The DSC data reveal a glass transition at around 65 °C, a cold crystallization peak at ~117 °C, and a melting temperature of approximately 153 °C. The thermal degradation, as indicated by the TGA, begins around 300 °C, affirming that PLA is thermally stable up to this point. The enthalpy of fusion and cold crystallization confirms moderate crystallinity, consistent with the XRD results.

## Author contributions

**Conceptualization:** Vishal V. Shukla, Yogesh G. Joshi.

**Formal analysis:** Vishal V. Shukla, Yogesh G. Joshi, Niraj Kumar Dewangan.

**Investigation:** Vishal V. Shukla, Yogesh G. Joshi.

**Methodology:** Yogesh G. Joshi.

**Software:** Vishal V. Shukla, Yogesh G. Joshi.

**Validation:** Vishal V. Shukla, Yogesh G. Joshi.

**Visualization:** Vishal V. Shukla, Yogesh G. Joshi, Niraj Kumar Dewangan.

**Writing – original draft:** Vishal V. Shukla, Yogesh G. Joshi.

**Writing – review & editing:** Niraj Kumar Dewangan.

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
