## [Decision Letter · Decision Letter 0]

26 Jan 2026

PONE-D-26-00919Development of a Real-Time Laser-Based Dimensional Monitoring System with Post-Mechanical Characterization of Synthesized Polylactic Acid 3D Prints in FDM Additive ManufacturingPLOS One

Dear Dr. Dewangan,

Thank you for submitting your manuscript to PLOS ONE. After careful consideration, we feel that it has merit but does not fully meet PLOS ONE’s publication criteria as it currently stands. Therefore, we invite you to submit a revised version of the manuscript that addresses the points raised during the review process.

We look forward to receiving your revised manuscript.

Kind regards,

Khalil Abdelrazek Khalil, Ph.D.

Academic Editor

PLOS One

**Journal Requirements:**

3. In the online submission form you indicate that your data is not available for proprietary reasons and have provided a contact point for accessing this data. Please note that your current contact point is a co-author on this manuscript. According to our Data Policy, the contact point must not be an author on the manuscript and must be an institutional contact, ideally not an individual. Please revise your data statement to a non-author institutional point of contact, such as a data access or ethics committee, and send this to us via return email. Please also include contact information for the third party organization, and please include the full citation of where the data can be found.

Reviewers' comments:

Reviewer's Responses to Questions

**Comments to the Author**

1. Is the manuscript technically sound, and do the data support the conclusions?

Reviewer #1: Yes

Reviewer #2: Yes

2. Has the statistical analysis been performed appropriately and rigorously? 

Reviewer #1: Yes

Reviewer #2: No

3. Have the authors made all data underlying the findings in their manuscript fully available?

The PLOS Data policy requires authors to make all data underlying the findings described in their manuscript fully available without restriction, with rare exception (please refer to the Data Availability Statement in the manuscript PDF file). The data should be provided as part of the manuscript or its supporting information, or deposited to a public repository. For example, in addition to summary statistics, the data points behind means, medians and variance measures should be available. If there are restrictions on publicly sharing data—e.g. participant privacy or use of data from a third party—those must be specified.requires authors to make all data underlying the findings described in their manuscript fully available without restriction, with rare exception (please refer to the Data Availability Statement in the manuscript PDF file). The data should be provided as part of the manuscript or its supporting information, or deposited to a public repository. For example, in addition to summary statistics, the data points behind means, medians and variance measures should be available. If there are restrictions on publicly sharing data—e.g. participant privacy or use of data from a third party—those must be specified.requires authors to make all data underlying the findings described in their manuscript fully available without restriction, with rare exception (please refer to the Data Availability Statement in the manuscript PDF file). The data should be provided as part of the manuscript or its supporting information, or deposited to a public repository. For example, in addition to summary statistics, the data points behind means, medians and variance measures should be available. If there are restrictions on publicly sharing data—e.g. participant privacy or use of data from a third party—those must be specified.requires authors to make all data underlying the findings described in their manuscript fully available without restriction, with rare exception (please refer to the Data Availability Statement in the manuscript PDF file). The data should be provided as part of the manuscript or its supporting information, or deposited to a public repository. For example, in addition to summary statistics, the data points behind means, medians and variance measures should be available. If there are restrictions on publicly sharing data—e.g. participant privacy or use of data from a third party—those must be specified.

Reviewer #1: Yes

Reviewer #2: Yes

4. Is the manuscript presented in an intelligible fashion and written in standard English?

Reviewer #1: Yes

Reviewer #2: Yes

5. Review Comments to the Author

Reviewer #1: 1. A direct comparison with previous methods is needed to show how the new Laser Dimension Measurement System (LDMS) is different from current laser scanning, profilometry, and in-situ monitoring techniques, as the paper discusses a key issue in FDM additive manufacturing.

2. The authors need to either provide timing metrics to back up their claim of "real-time" dimensional monitoring or change the terminology to reflect layer-wise or quasi-real-time monitoring since the suggested system uses a pause-scan-resume strategy.

3. Even though the manuscript suggests that it can detect and fix dimensional errors in real-time, there is no experimental proof showing that dimensions actually improve after using feedback, a closed-loop control strategy, or a correction algorithm.

4. Lacking information about the laser sensor's calibration procedure, measurement range, resolution, sampling frequency, and repeatability makes it impossible to evaluate the accuracy of the measurements and the reliability of the system.

5. Provide the number of samples, how consistent the results are, the standard deviation, and the confidence intervals to show reliability; the statistical evidence for comparing the LDMS with CMM and caliper measurements is not enough

6. The results of the dimensional accuracy test are only applicable to simple geometries; the system's capability to handle more complicated geometries, taller builds, and longer print times is still unknown.

7. There is no statistical analysis showing how pauses affect the thermal history and the strength properties, and there is no detailed comparison of traditional FDM with LDMS-assisted printing in terms of mechanical characteristics.

8. Even though people claim that PLA is amorphous, biodegradable, and performs well, there isn't enough detailed information about how it is made and tested, and it hasn't been compared to commercial PLA.

8. There are several theoretical assertions about automated dimensional comparison, point-cloud processing, and cloud-based CAD reconstruction that are unsubstantiated by evidence, including validation of workflows or performance metrics.

9. Significant language editing is necessary to remove informal expressions, grammatical mistakes, and repetitive phrasing from the manuscript. Additionally, the reveal improves the figures and references to meet the clarity and reproducibility standards.

Reviewer #2: Dear Author,

Your paper demonstrates a strong combination of experimental and testing work. A major revision is recommended. However, several questions have been raised for your consideration. Please address these points and revise the manuscript accordingly, which will enhance its value to the research community

1. How were pause-layer intervals selected, and how do they affect dimensional drift and print defects?

2. How was laser sensor accuracy validated dynamically during printer vibration and nozzle motion?

3. Why was ±1 mm sensor error considered acceptable for micro-scale FDM dimensional deviations?

4. How was thermal distortion of PLA isolated from extrusion and mechanical vibration errors?

5. How does the LDMS scanning time influence layer cooling, bonding, and residual stress formation?

6. What mechanism explains ±0.04 mm agreement with CMM despite sensor resolution limits?

7. How does point-cloud to CAD conversion uncertainty propagate into dimensional correction decisions?

8. Why do LDMS-printed parts show brittle tensile behavior despite intermittent thermal relaxation?

9. What microstructural changes link flexural ductility improvement over tensile brittleness?

10. How does layer-wise correction alter interlayer bonding strength compared with conventional FDM?

11. Why does LDMS improve dimensional accuracy but not significantly enhance mechanical strength?

12. How scalable is the LDMS system for continuous industrial printing without time or cost penalties?

13. The following articles are relevant to your field and should be cited in the paper.

Jannet, S., Soundararajan, R., Kandavalli, et al. Wear and Friction Behavior on Extrusion-Based 3D Printed Short Carbon Fiber Reinforced PETG Composites with Annealing and Laser Treated Effects. J. Inst. Eng. India Ser. D (2024). https://doi.org/10.1007/s40033-024-00794-6

M.S. Srinidhi, R. Soundararajan, K.S. Satishkumar, S. Suresh, Enhancing the FDM infill pattern outcomes of mechanical behavior for as-built and annealed PETG and CFPETG composites parts, Materials Today: Proceedings 45 (2021) 7208–7212, https://doi.org/10.1016/j.matpr.2021.02.417

K.SathishKumar, R. Soundararajan, G.Shanthosh, P.Saravanakumar, M.Ratteesh, “Augmenting Effect of Infill Density and Annealing on Mechanical Properties of PETG and CFPETG Composites Fabricated by FDM”, Materials Today Proceedings, https://doi.org/10.1016/j.matpr.2020.10.078.

Soundararajan Ranganathan, Sathish Kumar K, Shanthosh Gopal, and Ramprakash Palanivelu, “Enhancing the Tribological Properties PETG and CFPETG Composites Fabricated by FDM via Various Infill Density and Annealing,” SAE Technical Paper, 2020-28-0429, 2020, https://doi:10.4271/2020-28-0429

Soundararajan Ranganathan, Hari Nishok Rangasamy Suguna Thangaraj, Aravind Kumar Vasudeevan and Dharshan Karthick Shanmugam, “Analogy of Thermal Properties of Polyamide 6 Reinforced with Glass Fiber and Glass Beads through FDM Process”, SAE Technical Paper, 28(0137), pp: 6 Pages, 2019. https://doi.org/10.4271/2019-28-0137

6. PLOS authors have the option to publish the peer review history of their article (what does this mean?). If published, this will include your full peer review and any attached files.). If published, this will include your full peer review and any attached files.). If published, this will include your full peer review and any attached files.). If published, this will include your full peer review and any attached files.

...

Reviewer #1: No

Reviewer #2: No

---

## [Author Response · Author response to Decision Letter 1]

6 Mar 2026

Rebuttal Letter for PONE-D-26-00919

Christos Markides

Khalil Abdelrazek Khalil, Ph.D.

Academic Editor

PLOS One.

Thank you for considering the manuscript PONE-D-26-00919 (Research Paper: Development of

Real-Time Laser-Based Dimensional Monitoring System with Post-Mechanical Characterization of

Synthesised Polylactic Acid 3D Prints in FDM Additive Manufacturing) for publication in PLOS

one. We found the Reviewers comments very helpful and constructive.

We are sending the rebuttal letter explaining the changes performed on the manuscript. We have

addressed all the changes recommended by the reviewer and we are confident that the new version

of the manuscript is easier to understand and has a more fluent scientific discourse. The

revisions are addressed below.

Reviewer 1 Comments:

1. A direct comparison with previous methods is needed to show how the new Laser Dimension Measurement System (LDMS) is different from current laser scanning, profilometry, and in-situ monitoring techniques, as the paper discusses a key issue in FDM additive manufacturing.

Reponses: A dedicated Table 1 titled “Comparative Analysis of LDMS with Existing Monitoring Techniques” has now been added in the revised manuscript. This table includes a structured comparison highlighting the technical differences between the proposed LDMS and previously reported laser scanning, profilometry, and vision-based in-situ monitoring systems. Unlike earlier methods that primarily focus on surface profiling or defect detection, the LDMS integrates three-axis laser triangulation sensing, real-time point cloud to CAD conversion, direct G-code comparison, and closed-loop calibration capability through a pause–scan–resume strategy. These additions clarify the novelty and technical distinction of the proposed system.

2. The authors need to either provide timing metrics to back up their claim of "real-time" dimensional monitoring or change the terminology to reflect layer-wise or quasi-real-time monitoring since the suggested system uses a pause-scan-resume strategy.

Reponses: We sincerely thank the reviewer for this valuable observation. The proposed LDMS operates using a pause–scan–resume mechanism at predefined layer intervals. To ensure terminological precision, the manuscript has been revised to replace the term “real-time” with “layer-wise in-situ monitoring.” This more accurately reflects the operational strategy of the system, which performs dimensional verification during intermediate build stages rather than continuous real-time sensing. The terminology has been revised consistently in the Title, Abstract, Introduction, Section 4.2 (LDMS Complete Working), and Conclusion to reflect this clarification.

3. Even though the manuscript suggests that it can detect and fix dimensional errors in real-time, there is no experimental proof showing that dimensions actually improve after using feedback, a closed-loop control strategy, or a correction algorithm.

Response: We sincerely thank the reviewer for this important comment. The present study experimentally validates the dimensional monitoring capability of the LDMS through quantitative comparison with CMM measurements. A dedicated Table 2 has been added in the revised manuscript presenting nominal, CMM-measured, and LDMS-measured dimensions, demonstrating high agreement and confirming reliable deviation detection. While the LDMS architecture enables closed-loop correction, quantitative validation of automatic correction performance was not the primary objective of the present study. The manuscript has been revised to clarify this distinction and avoid overstatement. The relevant modifications have been highlighted.

4. Lacking information about the laser sensor's calibration procedure, measurement range, resolution, sampling frequency, and repeatability makes it impossible to evaluate the accuracy of the measurements and the reliability of the system.

Response: In the revised manuscript, a dedicated subsection titled “Laser Sensor Calibration and Measurement Specifications” has been added under Section 4 to provide comprehensive details regarding the metrological characteristics of the employed laser triangulation sensor. The added section includes: Sensor model and operational principle, Measurement range, Resolution and ADC limitations, Sampling frequency, Calibration methodology and Repeatability assessment. A summary table of the sensor specifications has also been incorporated to enhance clarity. These additions allow a more rigorous evaluation of the measurement accuracy and reliability of the proposed LDMS. All changes have been highlighted in the revised manuscript.

5. Provide the number of samples, standard deviation, confidence intervals, and statistical comparison to demonstrate reliability.

Response: We sincerely thank the reviewer for this valuable suggestion. In the revised manuscript, a dedicated subsection titled “Statistical Validation of Dimensional Measurements” has been added under Section 6.1. The number of samples (n), mean values, standard deviations, and 95% confidence intervals for LDMS, CMM, and digital caliper measurements have now been explicitly reported. These additions provide quantitative evidence of measurement consistency and system reliability. All changes have been highlighted in the revised manuscript.

6. The dimensional accuracy results are only applicable to simple geometries; the system's capability for complex geometries, taller builds, and longer print times is unknown.

Response: The present study focused on representative simple geometries to establish the baseline dimensional monitoring capability of the LDMS under controlled conditions. To clarify the scope of validation, additional statements have been incorporated into Section 6.1 and the Conclusion, which explicitly acknowledges that further investigation is required to evaluate performance for complex geometries, taller builds, and extended-duration printing. The scalability considerations and future validation directions have been highlighted in the revised manuscript.

7. There is no statistical analysis showing how pauses affect the thermal history and the strength properties, and there is no detailed comparison of traditional FDM with LDMS-assisted printing in terms of mechanical characteristics.

Response: We sincerely thank the reviewer for this valuable comment. The primary objective of the present work was to develop and validate the LDMS architecture for layer-wise dimensional monitoring and deviation detection. The pause–scan–resume mechanism was designed with minimal pause duration to avoid significant thermal disturbances during printing. To clarify the scope of the current study, additional statements have been incorporated in the Results and Discussion section explaining that a detailed statistical comparison between conventional FDM and LDMS-assisted printing, including the influence of pause intervals on thermal history and mechanical properties, is beyond the current scope and will be addressed in future investigations. These additions have been highlighted in the revised manuscript.

8. Even though people claim that PLA is amorphous, biodegradable, and performs well, there isn't enough detailed information about how it is made and tested, and it hasn't been compared to commercial PLA.

Response: Additional clarification regarding the PLA material used in this study has been incorporated in the Materials section. The source and characteristics of the commercially available PLA filament have been described, and it has been clarified that the mechanical and thermal behavior of the printed samples was evaluated through tensile, flexural, and thermal analyses. Furthermore, the observed material behavior has been discussed in relation to typical properties reported for commercial PLA in FDM applications. The corresponding revisions have been highlighted in the manuscript.

9. There are several theoretical assertions about automated dimensional comparison, point-cloud processing, and cloud-based CAD reconstruction that are unsubstantiated by evidence.

Response: The description of the dimensional comparison workflow has been clarified in Section 4.2 to distinguish between the implemented monitoring framework and conceptual extensions of the system. The text now explains that the current LDMS implementation performs layer-wise dimensional deviation detection using sensor measurements and reference CAD geometry, rather than full automated point-cloud reconstruction or cloud-based geometric correction. These clarifications ensure that the manuscript accurately reflects the scope of the implemented system. The corresponding revisions have been highlighted in the revised manuscript.

10. Significant language editing is necessary to remove informal expressions, grammatical mistakes, and repetitive phrasing from the manuscript. Additionally, the reveal improves the figures and references to meet the clarity and reproducibility standards.

Response: The manuscript has been carefully revised to improve overall language quality, remove informal expressions, correct grammatical errors, and eliminate repetitive phrasing. In addition, the figures have been refined to enhance clarity, and figure captions and in-text references have been revised to ensure better readability and reproducibility. All corresponding changes have been incorporated and highlighted in the revised manuscript.

Reviewer 2 Comments:

1. How were pause-layer intervals selected, and how do they affect dimensional drift and print defects?

Response: An additional clarification has been added in Section 4.2 regarding the selection of pause-layer intervals used in the LDMS-assisted printing workflow. The pauses were implemented at predefined layer checkpoints to enable dimensional verification while maintaining minimal interruption to the printing process. The pause duration was kept short to minimize potential thermal disturbances or interlayer bonding effects. Experimental observations indicated no noticeable dimensional drift or print defects associated with the pause mechanism under the tested conditions. The corresponding clarification has been highlighted in the revised manuscript.

2. How was laser sensor accuracy validated dynamically during printer vibration and nozzle motion?

Response: In the proposed LDMS framework, dimensional measurements are performed during the pause stage of the pause–scan–resume cycle, when the printer motion is temporarily halted. This approach allows the laser sensor to operate under quasi-static conditions, thereby minimizing the influence of printer vibrations and nozzle motion on measurement accuracy. The dimensional measurements obtained from the LDMS were subsequently validated through comparison with CMM measurements. A clarification explaining this measurement strategy has been added in Section 4.2 of the revised manuscript and highlighted for the reviewer’s convenience.

3. Why was ±1 mm sensor error considered acceptable for micro-scale FDM dimensional deviations?

Response: The laser triangulation sensor used in the LDMS has a nominal measurement uncertainty of approximately ±1 mm under practical operating conditions. However, the purpose of the proposed system is not to perform micron-level metrological inspection, but rather to detect significant dimensional deviations that may arise during the FDM printing process. Such deviations can accumulate over multiple layers due to extrusion inconsistencies, mechanical disturbances, or process drift, and are typically observable at the millimeter scale.

4. How was thermal distortion of PLA isolated from extrusion and mechanical vibration errors?

Response: The LDMS framework is designed to monitor the overall dimensional deviation of printed components during the fabrication process. The dimensional variations observed in the study therefore represent the combined influence of thermal contraction, extrusion variability, and mechanical disturbances during printing. The primary objective of the present work is the development of a monitoring framework for dimensional deviation detection rather than the isolation of individual error sources.

5. How does the LDMS scanning time influence layer cooling, bonding, and residual stress formation?

Response: In the proposed LDMS workflow, the scanning operation is performed during short pause intervals introduced at predefined layer checkpoints. The pause duration was intentionally kept minimal so that the deposited layer does not experience significant thermal cooling before printing resumes. Under the experimental conditions considered in this study, no visible defects or mechanical degradation attributable to the pause–scan process were observed, as reflected in the tensile and flexural test results. The primary focus of the present work is dimensional monitoring, while a detailed investigation of thermal history, interlayer bonding behavior, and residual stress development under varying pause durations will be considered in future studies.

6. What mechanism explains ±0.04 mm agreement with CMM despite sensor resolution limits?

Response: The ±0.04 mm value reported in the manuscript refers to the observed dimensional agreement between the printed components and the reference CMM measurements rather than the intrinsic resolution of the laser sensor itself. The LDMS measurements are used to estimate dimensional deviations during the layer-wise monitoring process, while the CMM provides high-precision post-process verification of the final printed geometry. The reported agreement therefore, reflects the overall dimensional consistency of the printed parts rather than the direct measurement resolution of the sensor. The LDMS is primarily intended for detecting practical dimensional drift during printing rather than replacing high-precision metrology instruments.

7. How does point-cloud to CAD conversion uncertainty propagate into dimensional correction decisions?

Response: In the present LDMS implementation, the point-cloud data generated during the scanning process are used primarily to estimate dimensional deviations relative to the reference CAD geometry. The system is designed for dimensional monitoring and deviation detection rather than fully automated geometric correction. Consequently, potential uncertainties associated with point-cloud acquisition and CAD comparison do not directly propagate into automatic correction decisions within the current workflow. Instead, the dimensional comparison provides an approximate indication of geometric deviation that can guide process monitoring and manual evaluation. Future work will investigate more advanced geometric reconstruction and uncertainty-aware correction algorithms to further improve automated dimensional compensation.

8. Why do LDMS-printed parts show brittle tensile behavior despite intermittent thermal relaxation?

Response: We thank the reviewer for this insightful question. The brittle tensile behavior observed in the printed specimens is primarily attributed to the intrinsic mechanical characteristics of PLA and the layer-by-layer deposition mechanism of the FDM process. PLA is known to exhibit relatively low elongation at break compared with more ductile thermoplastics, and FDM-printed parts often display anisotropic mechanical behavior governed by interlayer bonding strength. Although the LDMS workflow introduces short pause intervals for dimensional scanning, these pauses are brief and do not significantly alter the thermal history or polymer chain mobility within the deposited layers. Consequently, the overall fracture behavior remains consistent with the typical brittle response reported for PLA components fabricated using FDM.

9. What microstructural changes link flexural ductility improvement over tensile brittleness?.

Response: The difference between the tensile and flexural responses primarily arises from the distinct stress states involved in the two testing configurations rather than from significant microstructural changes in the PLA material. In tensile testing, the specimen experiences uniform tensile stress across the cross-section, making the failure strongly dependent on interlayer bonding strength, which often leads to brittle fracture in FDM-printed P

---

## [Decision Letter · Decision Letter 1]

17 Mar 2026

Development of a Layer-Wise In-Situ Laser-Based Dimensional Monitoring System with Post-Mechanical Characterization of Synthesized Polylactic Acid 3D Prints in FDM Additive Manufacturing

PONE-D-26-00919R1

Dear Dr. Dewangan,

We’re pleased to inform you that your manuscript has been judged scientifically suitable for publication and will be formally accepted for publication once it meets all outstanding technical requirements.

Kind regards,

Khalil Abdelrazek Khalil, Ph.D.

Academic Editor

PLOS One

Additional Editor Comments (optional):

Reviewers' comments:

Reviewer's Responses to Questions

**Comments to the Author**

1. If the authors have adequately addressed your comments raised in a previous round of review and you feel that this manuscript is now acceptable for publication, you may indicate that here to bypass the “Comments to the Author” section, enter your conflict of interest statement in the “Confidential to Editor” section, and submit your "Accept" recommendation.

Reviewer #1: All comments have been addressed

Reviewer #2: All comments have been addressed

2. Is the manuscript technically sound, and do the data support the conclusions?

Reviewer #1: Yes

Reviewer #2: Yes

3. Has the statistical analysis been performed appropriately and rigorously? 

Reviewer #1: Yes

Reviewer #2: Yes

4. Have the authors made all data underlying the findings in their manuscript fully available?

The PLOS Data policy requires authors to make all data underlying the findings described in their manuscript fully available without restriction, with rare exception (please refer to the Data Availability Statement in the manuscript PDF file). The data should be provided as part of the manuscript or its supporting information, or deposited to a public repository. For example, in addition to summary statistics, the data points behind means, medians and variance measures should be available. If there are restrictions on publicly sharing data—e.g. participant privacy or use of data from a third party—those must be specified.requires authors to make all data underlying the findings described in their manuscript fully available without restriction, with rare exception (please refer to the Data Availability Statement in the manuscript PDF file). The data should be provided as part of the manuscript or its supporting information, or deposited to a public repository. For example, in addition to summary statistics, the data points behind means, medians and variance measures should be available. If there are restrictions on publicly sharing data—e.g. participant privacy or use of data from a third party—those must be specified.requires authors to make all data underlying the findings described in their manuscript fully available without restriction, with rare exception (please refer to the Data Availability Statement in the manuscript PDF file). The data should be provided as part of the manuscript or its supporting information, or deposited to a public repository. For example, in addition to summary statistics, the data points behind means, medians and variance measures should be available. If there are restrictions on publicly sharing data—e.g. participant privacy or use of data from a third party—those must be specified.requires authors to make all data underlying the findings described in their manuscript fully available without restriction, with rare exception (please refer to the Data Availability Statement in the manuscript PDF file). The data should be provided as part of the manuscript or its supporting information, or deposited to a public repository. For example, in addition to summary statistics, the data points behind means, medians and variance measures should be available. If there are restrictions on publicly sharing data—e.g. participant privacy or use of data from a third party—those must be specified.

Reviewer #1: Yes

Reviewer #2: Yes

5. Is the manuscript presented in an intelligible fashion and written in standard English?

Reviewer #1: Yes

Reviewer #2: Yes

6. Review Comments to the Author

Reviewer #1: (No Response)

Reviewer #2: Dear Authors,

The responses to all the reviewers’ queries have been addressed satisfactorily, and the overall quality of the manuscript is good. No further clarifications are required. I agree to accept this article for publication.

7. PLOS authors have the option to publish the peer review history of their article (what does this mean?). If published, this will include your full peer review and any attached files.). If published, this will include your full peer review and any attached files.). If published, this will include your full peer review and any attached files.). If published, this will include your full peer review and any attached files.

...

Reviewer #1: No

Reviewer #2: No

---

## [Editor Report · Acceptance letter]

PONE-D-26-00919R1

PLOS One

Dear Dr. Dewangan,

I'm pleased to inform you that your manuscript has been deemed suitable for publication in PLOS One. Congratulations! Your manuscript is now being handed over to our production team.

Kind regards,

on behalf of

Dr. Khalil Abdelrazek Khalil

Academic Editor

PLOS One